# Sexual Health in Patients with Atopic Dermatitis: A Cross-Sectional Study

**DOI:** 10.3390/medicina61101782

**Published:** 2025-10-02

**Authors:** Natalia Juśko, Magdalena Masajada, Anna Żabówka, Adam Ćmiel, Paweł Brzewski, Adam Reich

**Affiliations:** 1Department of Dermatology and Venereology, Stefan Zeromski Municipal Hospital, 31-913 Kraków, Poland; nataliajusko@yahoo.com (N.J.); brzewski@gmail.com (P.B.); 2Department of Applied Mathematics, AGH University of Science and Technology, 30-059 Kraków, Poland; cmiel@agh.edu.pl; 3Faculty of Medicine and Health Sciences, Andrzej Frycz Modrzewski Krakow University, 30-705 Kraków, Poland; 4Department of Dermatology, Faculty of Medicine, Medical College of Rzeszów University, 35-055 Rzeszów, Poland

**Keywords:** atopic dermatitis, quality of life, sexual life, well-being

## Abstract

*Background and objectives:* Atopic dermatitis (AD) is a chronic inflammatory skin disease that affects not only physical health but also psychological well-being. While the emotional and social burden of AD is well documented, there is still limited research on how AD affects sexual health. The study aimed to evaluate quality of life (QoL), mental health, and risk factors for impaired sexual life, as well as their relationships. *Materials and Methods:* A total of 201 participants (96 patients with AD and 105 healthy controls) were enrolled in the study. Socio-demographic and clinical data were obtained using a specifically developed questionnaire. In addition, participants completed validated scales, including the DLQI, HADS, FSFI, IIEF-5, and SRSLQ. AD severity was assessed using the SCORAD questionnaire. *Results:* Our study found that patients with AD had statistically significantly higher mean anxiety (6.8 ± 3.6 vs. 5.0 ± 3.2; *p* < 0.001), depression (5.2 ± 3.4 vs. 3.9 ± 2.9, *p* < 0.01), and skin-related sexual dysfunction scores (15.0 ± 4.5 vs. 4.4 ± 4.7, *p* < 0.001), as well as QoL scores (12.3 ± 6.1 vs. 1.8 ± 3.1, *p* < 0.001), than healthy controls. Female AD patients reported higher values of depression and anxiety compared to male patients (5.9 ± 3.1 vs. 4.4 ± 3.5, *p* = 0.03, 7.6 ± 2.9 vs. 6.0 ± 4.1, *p* = 0.03, respectively) and lower FSFI scores compared to healthy women (24.8 ± 8.0 vs. 31.3 ± 3.0, *p* < 0.001). Deterioration in sexual health, assessed by the SRSLQ score, was strongly correlated with QoL impairment (R = 0.5, *p* < 0.001), anxiety (R = 0.51, *p* < 0.001), and depression (R = 0.5, *p* < 0.001). Finally, we found that sex life negatively correlates with AD severity (*p*=0.001), involvement of a genital area (*p* = 0.005), intensity of pruritus (r = 0.284, *p* = 0.005), and insomnia (r = 0.366, *p* < 0.001). *Conclusions:* AD significantly affects patients’ quality of life, including their sex life. Many factors associated with the disease also contribute to the deterioration of patients’ sexual health. Routine assessment of sexual life in dermatological practice, using validated tools, could facilitate early identification and support for affected patients. *Significance:* This study highlights the often-overlooked impact of atopic dermatitis on patients’ sexual health. Our findings demonstrate that sexual function is significantly impaired in individuals with atopic dermatitis—particularly among women—and that such dysfunction is closely associated with disease-related symptoms. These results have important implications for improving the quality of care provided to individuals affected by the condition.

## 1. Introduction

Atopic dermatitis (AD) is one of the most common chronic inflammatory skin disease, whose prevalence has been increasing in recent years worldwide [1]. The epidemiological data varies by country, but the disease may affect approximately 20% of children and up to 14% of the adult population [2]. The steady increase in the incidence of the disease among adults may be explained by high persistence or adult onset of AD, making it not only a clinical but also a social problem.

It was documented that AD has a significant impact on patients’ quality of life (QoL) [1,3]. Recent studies showed that AD has the highest disability-adjusted life-year (DALY) burden among all skin diseases [4]. The patient burden relates directly to the physical signs (eczematous-like skin lesions of a specific location, depending on the patient’s age) and subjective symptoms of AD. Multiple factors generate social and psychological burden that patients with AD have to face daily. Over 90% of patients experience chronic itch and cutaneous pain [5], which may contribute to sleep disturbances reported in 33–90% of adult patients [6]. Studies have shown that AD patients who worry more about sleep deprivation have difficulties in falling and staying asleep, which interferes with their daily functioning. Furthermore, there was a strong association of AD severity with missed workdays, a higher number of physician visits, and overall poor health [7,8]. It is known that compared to healthy controls, patients with AD are at higher risk of developing various comorbidities, not only physical but also neuropsychiatric ones, such as anxiety and depression. Psychological distress may be additionally compounded by skin lesions, especially when visible, leading to decreased self-esteem. Ring et al. reported that more than half of all AD patients reported to be embarrassed about their skin and indicated AD to influence their choice of clothing [3]. Genital involvement was particularly perceived by patients as the most annoying location [9]. Including all of these factors, it is no wonder that AD takes a toll on sexual health and romantic relationships as well.

The impact of AD on sexual functions seems to be substantial, and it can result in significant alterations in QoL. A multicenter study involving 3485 patients with chronic dermatoses showed that almost one third of patients with AD declared a significant impact on sexual life, but, on the other hand, 96% of patients said that this problem had never been raised during medical consultation [10,11]. Some studies have linked AD with erectile dysfunction in men and decreased sexual satisfaction in women [12,13].

Since sexuality is an important aspect of life, and only a few studies have focused on this subject, we believe that more research should be conducted. The present study evaluates and compares sexual function, QoL, depression, and anxiety in patients with AD of both sexes in comparison to the healthy population. 

## 2. Materials and Methods

### 2.1. Study Design

This was an observational, cross-sectional, questionnaire-based study (Appendix A). The study was approved by a local Ethics Committee at Andrzej Frycz Modrzewski Krakow University (reference number KBKA/22/O/2022). As mentioned above, the aim of this study was to assess sexual function, QoL, depression, and anxiety in patients with AD of both sexes in comparison to the healthy population. All patients agreed to participate and signed the written informed consent form.

All patients underwent a thorough physical examination and were asked to complete a questionnaire regarding socio-demographic and personal information, including age, sex, family medical history, and previous treatment. Severity of the disease was assessed based on SCORAD score, with subsequent dividing of patients into mild/moderate/severe AD (i.e., mild AD if SCORAD was <25 points, moderate AD if SCORAD was ≥25 points and ≤50 points, and severe AD if SCORAD was >50 points) [14]. Next, all patients were asked to complete several questionnaires (see below)—before completing them, each participant received detailed explanations from the physician conducting the study (N.J., M.M., or A.Ż.) on how to correctly complete the tools used in the study.

### 2.2. Assessment of Sexual Dysfunction

To evaluate sexual dysfunction (SD), the Female Sexual Function Index (FSFI) [15] and the 5-Item International Index of Erectile Function (IIEF-5) was used for women and men, respectively [16]. In addition, the 11-item Skin-Related Sexual Life Questionnaire (SRSLQ) was employed for both sexes [17]. The FSFI consists of 19 questions divided into 6 domains: desire, arousal, lubrication, orgasm, sexual satisfaction, and pain. Questions are awarded from 0 to 5 points, and clinically significant female sexual dysfunction is diagnosed at values lower than or equal to 26 points. The IIEF-5 is a 5-item questionnaire in which questions are scored from 0 to 5, and male sexual dysfunction is identified when a score of 21 or less is obtained. SRLSQ is a recently validated tool for the assessment of possible sexual dysfunction and related psychological burdens in patients with skin diseases [17]. This questionnaire is divided into sections addressing psychosocial and emotional problems. The maximum sum that each patient may obtain is 40 points, while the minimum is 0.

### 2.3. Assessment of Quality of Life, Anxiety and Depression

The QoL was evaluated using the validated Polish language versions of the Dermatology Life Quality Index (DLQI) [18]. According to the scale, health-related QoL is considered minimally impaired with a score of 2 to 5, moderately impaired with a score of 6 to 10, very much impaired with a score of 11 and above, and extremely impaired with a score of 21 or greater. In addition, the Hospital Anxiety and Depression Scale (HADS) was used to evaluate the level of anxiety and depression [19]. It includes 7 items assessing anxiety and 7 assessing depression, each with 4 possible answers. For each dimension of anxiety and depression, a score from 0 to 7 is considered normal, from 8 to 10 borderline, and equal to or above 11 points indicates a case in need of further examination or treatment. All questionnaires were completed always in the same order: DLQI, HADS, SRLSQ, FSFI (women), or IIEF-5 (men).

### 2.4. Study Population

Participants were consecutively recruited between June 2022 and March 2024 from patients presenting for routine follow-up visits to the outpatient Dermatology Clinic and Dermatology–Allergology Clinic, as well as from those hospitalized for an exacerbation of disease at the Department of Dermatology, Stefan Żeromski Specialist Hospital in Krakow, Poland. Inclusion criteria included the following: male and female patients diagnosed with AD on the basis of the Hanifin and Rajka criteria, aged above 18 years [20]. Exclusion criteria were as follows: gynecological, urological, or autoimmune disorders potentially affecting sexual function, pregnancy, and other chronic and/or extensive skin disorders involving the genitals (psoriasis, bullous dermatoses, etc.) that might interfere with the current study aim. Before entering the study, patients had to be off any topical therapy except emollients for at least 2 weeks and could not use any systemic anti-inflammatory therapy for at least 4 weeks. The control group consisted of age- and gender-matched adults referred for evaluation of nevi at the outpatient Dermatology Clinic. Participants who declined to participate (n = 6 in the AD group; n = 8 in the control group), withdrew during the course of the study (n = 7 in the AD group; n = 12 in the control group), or failed to provide complete informed consent (*n* = 1 in the control group) were excluded from the final analysis. A total of 201 participants were enrolled, presented as follows: 96 patients with AD (47 males and 49 females) and 105 healthy controls (50 males and 55 females). Among them, some missing data were noted in the completion of self-reported questionnaires. In the AD group, one participant did not complete the HADS, three did not complete the IIEF, and two did not complete the FSFI. In the control group, missing data were recorded for the HADS (n = 2), IIEF (n = 2), FSFI (n = 2), and SRSLQ (n = 3).

### 2.5. Statistical Analysis

Statistical analysis was performed with Statistica 13.3 (TIBCO Software Inc., Kraków, Poland). Descriptive statistics were presented as mean ± standard deviation for continuous variables and the frequency for categorical variables. Associations between qualitative variables were analyzed using the chi-square test and the Cochran–Armitage test for trend. The General Linear Model (GLM) was used for group comparisons. Correlation between variables was estimated with the Spearman rank order correlations. A *p*-value ≤ 0.05 was be considered statistically significant.

## 3. Results

The mean age of the patients with AD was 36.3 ± 14.3 years, and that of the controls was 39.2 ± 11.2 years. There were no significant differences between the groups with regard to gender (χ^2^ = 0.02, *p* = 0.88) or age (t = 0.27, *p* = 0.113). Duration of AD ranged from 6 months to 70 years, with a mean of 22.4 ± 15.2 years. A total of 34.4% of patients presented skin lesions in the anogenital area. Based on the SCORAD index, 18 (18.7%) patients had a mild form, 45 (46.9%) had a moderate form, and 33 (34.4%) had a severe form of AD.

As shown in Table 1, patients with AD presented statistically significantly higher mean anxiety (AD patients: 6.8 ± 3.6 vs. healthy controls: 5.0 ± 3.2; *p* < 0.001) and depression scores (AD patients: 5.2 ± 3.4 vs. healthy controls: 3.9 ± 2.9, *p* < 0.01) and lower QoL scores than the healthy controls (12.3 ± 6.1 vs. 1.8 ± 3.1, respectively, *p* < 0.001) (Table 1). Furthermore, the evaluation of sexual problems revealed that FSFI scores for female AD patients were statistically lower compared to healthy women (24.8 ± 8.0 vs. 31.3 ± 3.0, *p* < 0.001). The IIEF-5 scores for male AD patients were slightly lower, but the difference compared to the control group was not statistically significant (21.3 ± 3.9 vs. 22.1 ± 4.3, respectively, *p* = 0.38).

**Table 1 medicina-61-01782-t001:** Comparison of patients with atopic dermatitis (AD) with healthy controls regarding quality of life, anxiety, depression, and sexual dysfunction (DLQI—the Dermatology Life Quality Index, HADS—the Hospital Anxiety and Depression Scale, FSFI—the Female Sexual Function Index, IIEF-5—the 5-item International Index of Erectile Function, SRSLQ—Skin-Related Sexual Life Questionnaire).

	Patients with AD	Controls	*p*-Value	Cohen’s d with 95%CI
DLQI	12.3 ± 6.1	1.8 ± 3.1	<0.001	2.20 [1.84; 2.5]
HADS Anxiety	6.8 ± 3.6	5.0 ± 3.2	<0.001	0.53 [0.24; 0.81]
HADS Depression	5.2 ± 3.4	3.9 ± 2.9	<0.01	0.42 [0.13; 0.69]
IIEF-5	21.3 ± 3.9	22.1 ± 4.3	0.38	−0.18 [−0.59; 0.23]
FSFI	24.8 ± 8.0	31.3 ± 3.0	<0.001	−1.11 [−1.52; 0.68]
SRSLQ	15.0 ± 4.5	4.4 ± 4.7	<0.001	1.69 [1.36; 2.01]

Female AD patients reported higher values of depression and anxiety scores compared to male patients (5.9 ± 3.1 vs. 4.4 ± 3.5, *p* = 0.03, Cohen’s d = 0.46 [95%CI: 0.06; 0.87], 7.6 ± 2.9 vs. 6.0 ± 4.1, *p* = 0.03, Cohen’s d = 0.46 [0.05; 0.86], respectively). No other statistically significant differences were noted between men and women with AD, as shown in Table 2.

**Table 2 medicina-61-01782-t002:** Comparison of males and females suffering from atopic dermatitis (AD) regarding psychosocial well-being (DLQI—the Dermatology Life Quality Index, HADS—the Hospital Anxiety and Depression Scale, SRSLQ—Skin-Related Sexual Life Questionnaire, N/A—not applicable).

	Men with AD	Women with AD	*p*-Value	Cohen’s d with 95%CI
DLQI	12.2 ± 6.6	12.4 ± 5.6	0.85	N/A
HADS Anxiety	6.0 ± 4.1	7.6 ± 2.9	0.03	0.46 [0.05; 0.86]
HADS Depression	4.4 ± 3.5	5.9 ± 3.1	0.03	0.46 [0.06; 0.87]
SRSLQ	13.8 ± 8.4	16.1 ± 6.4	0.14	N/A

Skin-related sexual dysfunction scores were greater in AD patients compared to controls (15.0 ± 4.5 vs. 4.4 ± 4.7, *p* < 0.001) (Table 1). Of the AD patients who completed the survey, more than 95% admitted that they felt at least slightly unattractive due to AD, and nearly 80% of surveyed participants stated that skin complaints at least occasionally affected their sex life. More than 70% of the responders said they felt ashamed at least occasionally when they were with their sexual partner. and nearly 40% admitted avoiding sexual contact often or even all the time. However, at the same time, almost 60% of the participants never experienced rejection due to their disease. Among other questions, more than 90% of participants stated that they feared that other people perceived their disease as contagious and at least occasionally felt embarrassed when skin lesions were located on visible body areas and in genital areas. Interestingly, female patients felt more unattractive, more embarrassed when skin lesions involved visible body areas and more often avoided sexual contact due to AD compared to male patients (*p* = 0.03; *p* = 0.01; *p* = 0.02, respectively). No statistically significant differences were observed for other questions between female and male patients. Furthermore, statistically significant positive correlations were observed between SD and QoL impairment (R = 0.52, *p* < 0.001), anxiety (R = 0.52, *p* < 0.001), depression (R = 0.51, *p* < 0.001), and negative correlation with sexual health in men (R = −0.45, *p* < 0.001) (Table 3). Moreover, sexual functioning was related to the objective AD severity (*p* = 0.001). Specifically, patients with severe AD had significantly higher skin-related sexual impairment (SRSLQ: 18.24 ± 1.24 points) compared to patients with mild AD (SRSLQ: 11.17 ± 1.67 points) (*p* = 0.01), and the difference between patients with moderate AD (SRSLQ: 14.13 ± 1.06 points) and severe AD was marginally significant (*p* = 0.054). In addition, relationships between SRSLQ and AD involvement of a sensitive area (*p* = 0.005), intensity of pruritus (r = 0.284, *p* = 0.005), and insomnia (r = 0.366, *p* < 0.001) were found. The other parameters studied had no effect on the measured SRSLQ level (Table 3 and Table 4).

All studied patients reported that AD had affected their QoL to some degree with the mean DLQI score of 12.3 ± 6.1 points (Figure 1). Importantly, 21.9% reported a moderate effect (score 6–10), 51% reported a very large effect (score 11–20), 10.4% reported an extremely large effect (score 21–30), and only 16.7% reported that AD had a small effect on their QoL (score 2–5). Statistically significant correlations were observed between QoL impairment and anxiety (r = 0.402, *p* < 0.001), depression (r = 0.38, *p* < 0.001), pruritus (r = 0.44, *p* < 0.001), skin-related sexual dysfunction (r = 0.498, *p* < 0.001), severity of the disease (r = 0.46, *p* < 0.001), and in men also with sexual health problems (r = 0.375, *p* = 0.012).

## 4. Discussion

Sexuality represents an expression of well-being, encompassing physical, emotional, mental, and social aspects [21]. The chronic and visible nature of AD can understandably have far-reaching consequences for sexual health and intimate relationships, as indicated by some studies to date [9,11,22,23]. In line with previous observations, the present study has demonstrated that patients with AD had lower QoL and a higher risk of depression in the healthy population. These patients also displayed worse sexual functioning, particularly in relation to skin-related complaints. Sexual dysfunction (SD) was significantly associated with both depression and anxiety and directly affected QoL. Among the independent factors directly related to the AD severity, involvement of a genital area, itching, and insomnia were negatively correlated with sexual health. On the other hand, the age of the studied population and the duration of AD had no effect on QoL. Finally, female patients showed higher levels of depression and anxiety, were more concerned about their attractiveness, especially when skin lesions involved visible areas, and more often avoided sexual contact due to the disease compared to male patients. Although clinical sexual dysfunction appeared more prevalent in men with atopic dermatitis compared to the control group, this difference was not statistically significant, whereas in women the difference proved to be statistically significant. This may indicate that the disease has a greater impact on women’s self-perception, mental health, and sexual well-being than on men’s.

The first studies on sexual health in AD date back to 1973 [24]. In a population of 1972 adults, 19% of patients reported that eczema affected their sex life. Another study conducted in France on 266 patients [23] reported a disease-related decrease in sexual desire in 57.5% of patients and 32% of their partners. Many subsequent studies have focused on the significant mental and emotional burden of the disease on individuals. Ring et al. [3] showed that over half of 1189 adult patients reported that AD had a moderate to extremely large effect on their QoL, and over 10% of those with moderate to severe AD exhibited depressive symptoms. Schonmann et al. [25] found that AD was associated with a 14% increase in the risk of newly diagnosed depression and a 17% increase in the risk of a subsequent anxiety diagnosis. Finally, Sampogna et al. [11] indicated a strong association between both depression and anxiety and impairments in sexual life. Our study yields similar observations, as the risk of depression and anxiety was shown to be higher and the QoL decreased among AD patients compared to the healthy population. Moreover, a significant correlation between these parameters and sexual dysfunction indicates the interference of AD and its mental comorbidities with sexual function which can have profound implications for patients’ QoL.

The psychosocial burden of AD is often attributed to reduced self-esteem issues resulting from altered skin appearance, especially when lesions occur in visible areas of the body. According to two French studies, involvement of visible body areas negatively correlates with romantic relationships, sexual health, and QoL [9,22]. Among other locations, genital involvement was found to have the strongest association with experiencing sexual dysfunction [26,27]. Patients with genital involvement reported significantly higher disease burden and lower QoL than patients without genital lesions [9]. In another study, almost half of surveyed patients reported genital eczema at some point during the course of their disease, and 75.75% of them reported that their genital lesions interfered with their sexual life [28], and female patients appear to be significantly more affected than men [28,29]. These observations were consistent with our results and emphasize the need for specific management of AD with respect to lesions, as well as careful inspection of the genital area in every patient. In addition, we cannot exclude the possibility that depressive mood observed in many patients with AD may further contribute to lower self-esteem, stigmatization, and problems in sexual life.

Sleep disturbances, as well as the presence of itching and pain, are very common and burdensome in AD [5,6]. Their influence on quality of life has been extensively explored, but studies focusing on their relation to sexual health are lacking. In a multicenter European study assessing several skin conditions, AD was shown to have a particularly high impact on sexual difficulties, and this impact was strongly associated with itching [11]. Misery et al. [23] found that both patients and their partners experienced excessive daytime sleepiness, which worsened AD severity. Finally, Kaundinya et al. [27] found that pruritus and sleep impairment were directly associated with sexual dysfunction. In the current study, both itching and sleeplessness, assessed with SCORAD’s analog scales, were negatively correlated with sexual health. Having this in mind, SCORAD proves to be a helpful tool for identifying patients who may potentially experience sexual problems [9,27]. Patients with severe AD are significantly more likely to report a decrease in libido due to AD compared to those with mild or moderate forms of the disease [30].

This study is one of the few to provide a gender-inclusive assessment of sexual dysfunction in patients with atopic dermatitis compared to healthy controls. However, its findings should be interpreted with caution due to limitations such as a small sample size, single-center design, and reliance on self-reported data, which may affect the accuracy of the results. In addition, the quantification of the results obtained with the questionnaires may be imprecise; therefore, this research cannot pertinently indicate the causal relationship between AD and sexual dysfunction. Some drugs used by the patients in the past might also contribute to sexual dysfunction, and this aspect was not considered in the current study. Furthermore, another limiting factor is the lack of a psychiatrist in the research team who could confirm the diagnosis of depression or anxiety. However, as previous studies have shown similar findings [9,11,22,23], we do believe that our results are valid, although further research involving larger, more diverse populations and objective measures of sexual health is needed to confirm and expand on these results.

## 5. Conclusions

To improve patients’ QoL in routine care, the current extent of AD, the location of skin lesions, subjective symptoms, sleep disturbances, and comorbidities should be assessed to identify potential deficits in patients’ sexual health. A more insightful approach to female patients may be needed. However, the instruments used to assess sexual dysfunction have varied widely to date. The most commonly used measure was the nineth question of the DLQI, but this is not a validated scale for this specific purpose. The SRSLQ, on the other hand, has been validated as a stand-alone measure of sexual dysfunction and may be useful for further research and in routine practice to assess sexual dysfunction secondary to skin disease.

## Figures and Tables

**Figure 1 medicina-61-01782-f001:**
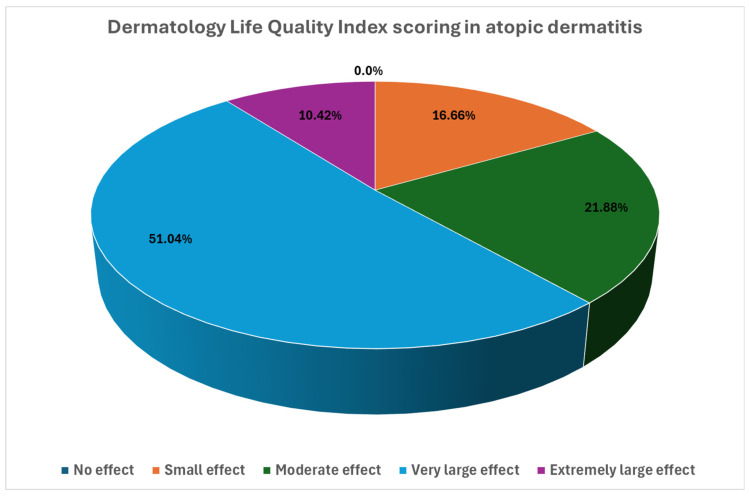
The effect of AD on the patients’ quality of life.

**Table 3 medicina-61-01782-t003:** Correlation between the level of sexual dysfunction assessed with SRSLQ (Skin-Related Sexual Life Questionnaire) and other studied parameters (DLQI—the Dermatology Life Quality Index, FSFI—the Female Sexual Function Index, HADS—the Hospital Anxiety and Depression Scale, IIEF-5—the 5-item International Index of Erectile Function, SCORAD—Scoring of Atopic Dermatitis).

	SRSLQ
	Pearson’s Correlation
Age (years)	r = 0.09, *p* = 0.36
Disease duration (years)	r = −0.09, *p* = 0.39
SCORAD (points)	r = 0.33, *p* = 0.001
Itch intensity (points)	r = 0.28, *p* = 0.005
Sleeping problems (points)	r = 0.37, *p* < 0.001
DLQI (points)	r = 0.52, *p* < 0.001
HADS Anxiety (points)	r = 0.52, *p* < 0.001
HADS Depression (points)	r = 0.51, *p* < 0.001
IIEF-5 (points)	r = −0.45, *p* < 0.001
FSFI (points)	r = −0.26, *p* = 0.07

**Table 4 medicina-61-01782-t004:** Relationship between the level of sexual dysfunction assessed with SRSLQ (Skin-Related Sexual Life Questionnaire) and other studied parameters (SCORAD—Scoring of Atopic Dermatitis).

		n	SRSLQ (Points)	*p*-Values
			Mean ± SD	
Gender	Males	47	13.83 ± 8.42	*p* = 0.141
	Females	49	16.10 ± 6.38	
Disease severity acc. to SCORAD	Mild atopic dermatitis	18	11.17 ± 1.67	*p =* 0.0025
	Moderate atopic dermatitis	45	14.13 ± 1.06	
	Severe atopic dermatitis	33	18.24 ± 1.24	
Employment	Employed	60	15.12 ± 0.97	*p* = 0.37
	Unemployed	9	17.11 ± 2.5	
	Student	17	12.47 ± 1.82	
	Retired	10	16.6 ± 2.37	
Education	Primary school	3	13.33 ± 4.64	*p* = 0.675
	Secondary school	13	15.0 ± 12.14	
	High school	34	14.06 ± 1.32	
	University	39	16.23 ± 1.24	
Marital status	Married	43	14.79 ± 1.11	*p* = 0.9
	Single	43	15.19 ± 1.16	
	Divorced	8	15.88 ± 2.68	
	Widower	2	11.5 ± 5.37	
Atopic lesions within genital area	Yes	33	18.09 ± 7.78	*p* = 0.005
	No	62	13.42 ± 6.91	

## Data Availability

Data are available from the corresponding author upon reasonable request.

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
