# Peer review of "Sexual Health in Patients with Atopic Dermatitis: A Cross-Sectional Study"

_medicina, 2025, doi:10.3390/medicina61101782_

Round 1

Reviewer 1 Report

Comments and Suggestions for Authors

This is an article about the effect of atopic dermatitis on sexual health. The topic is clinically relevant, but the manuscript’s contribution lacks novelty beyond prior work on AD, quality of life, and sexual health.

There are some important notes to take into account:

  1. Text claims “clinical sexual dysfunction was more prevalent in men with AD vs controls,” yet IIEF-5 shows no significant difference (p=0.38)
  2. "Patients with severe AD (SCORAD: 18.24±1.24 points)". The value shown in Table 4 is the SRSLQ, not SCORAD - please correct accordingly

  3. In Table 4 “Psoriatic lesions within genital area” is probably meant to be atopic lesions. Please check if it is correct

  4. The phrase “involvement of a sensual area” is not standard clinical terminology, please use genital or sensitive area instead

  5. Please report effect sizes (Cohen’s d or Hedges g) and 95% CIs for all group comparisons, not just p-values

  6. Please add detailed inclusion/exclusion, recruitment setting, and control selection/matching. Moreover, it would be interesting to show a baseline table (age, sex, BMI, comorbidities, therapies, disease duration) to assess confounding

Author Response

Reviewer: 

This is an article about the effect of atopic dermatitis on sexual health. The topic is clinically relevant, but the manuscript’s contribution lacks novelty beyond prior work on AD, quality of life, and sexual health.

There are some important notes to take into account:

Text claims “clinical sexual dysfunction was more prevalent in men with AD vs controls,” yet IIEF-5 shows no significant difference (p=0.38)

Authors: The sentence has been changed as follows: 

The IIEF-5 scores for male AD patients were slightly lower, but the difference compared to the control group was not statistically significant (21.3±3.9 vs 22.1±4.3, respectively, p=0.38). 

Reviewer: "Patients with severe AD (SCORAD: 18.24±1.24 points)". The value shown in Table 4 is the SRSLQ, not SCORAD - please correct accordingly.

Authors: Patients with severe AD were defined as having SCORAD >50 points. The values in brackets refer to SRSLQ, not to SCORAD. The mistake was corrected. 

Reviewer: In Table 4 “Psoriatic lesions within genital area” is probably meant to be atopic lesions. Please check if it is correct

Authors: We thank the reviewer for pointing this out. The mistake has been corrected. 

Reviewer: The phrase “involvement of a sensual area” is not standard clinical terminology, please use genital or sensitive area instead

Authors: The phrase has been changed as suggested. 

Reviewer: Please report effect sizes (Cohen’s d or Hedges g) and 95% CIs for all group comparisons, not just p-values

Authors: We have provided Cohen's d for all significantly different values. 

Reviewer: Please add detailed inclusion/exclusion, recruitment setting, and control selection/matching. Moreover, it would be interesting to show a baseline table (age, sex, BMI, comorbidities, therapies, disease duration) to assess confounding.

Authors: We have provided the inclusion and exclusion criteria in the Method section. 

Reviewer 2 Report

Comments and Suggestions for Authors

Dear authors,

The manuscript titled, “Sexual Health in Patients with Atopic Dermatitis: A Cross-Sectional Study”, evaluated and compares sexual function, QoL, depression and anxiety in patients with AD of both sexes in comparison to healthy population. The results indicated that AD has a substantial impact on the quality of life of patients, including their sexual life.  The disease is associated with numerous factors that contribute to the decline of patients' sexual health. Moreover, women are more affected by the disease than men are in terms of their sexual well-being, mental health, and self-perception.

The manuscript is relevant and provides an overview of Sexual Health in Patients with Atopic Dermatitis.

In order to better analyze and comprehend the patients' responses, I suggest that the study's questions be included in the supplementary files.

Author Response

Reviewer: The manuscript titled, “Sexual Health in Patients with Atopic Dermatitis: A Cross-Sectional Study”, evaluated and compared sexual function, QoL, depression, and anxiety in patients with AD of both sexes in comparison to the healthy population. The results indicated that AD has a substantial impact on the quality of life of patients, including their sexual life.  The disease is associated with numerous factors that contribute to the decline of patients' sexual health. Moreover, women are more affected by the disease than men are in terms of their sexual well-being, mental health, and self-perception.

The manuscript is relevant and provides an overview of Sexual Health in Patients with Atopic Dermatitis.

In order to better analyze and comprehend the patients' responses, I suggest that the study's questions be included in the supplementary files.

Authors: We have provided study aims in the main text (Study design subchapter). 

Reviewer 3 Report

Comments and Suggestions for Authors

Starting from the ultrastructural aspect of keratinocyte epithelial cells, intestinal epithelial cells, and the blood-brain barrier, the presence of the same tight apical junctional pattern, formed by membrane glycoproteins such as occludins and claudins, is highlighted. The expression of these molecules is controlled by a series of endogenous and exogenous factors, with repercussions on systemic homeostasis. Moreover, this morphological feature justifies and supports the involvement of the brain-skin-gut axis in the etiopathogenesis and clinical evolution of atopic dermatitis, one of the most common chronic immune-mediated inflammatory skin diseases.

To date, it is unclear whether anxiety and depression can be "unmasked" using skin keratinocytes or whether keratinocyte damage can attract and amplify depression/anxiety. Given the individual and social psycho-emotional impact of this pathology, the concern of the authors of this research is commendable. Therefore, the Introduction chapter could be supplemented with a table indicating the role of the main predisposing and/or contributing factors to sexual dysfunction in the context of atopic dermatitis.

Completing and using self-report questionnaires requires careful and thorough training by health professionals. Therefore, the Materials and Methods section must be completed with clear information on how to apply the various instruments used.

There are several degrees of severity of depression, which cannot be accurately assessed using a scale alone. Several scales are proposed in clinical practice—Hamilton, Beck, etc. On the other hand, the certification of data and the correct and adapted interpretation of results require the intervention of a specialist (psychologist and/or psychiatrist). It is not clear from the article whether this study benefited from the intervention of such a specialist.

Patients with gynecological/urological pathologies were excluded, but nothing is specified about the presence/absence of other chronic immune-mediated inflammatory pathologies such as autoimmune thyroiditis. It is known that females experience variations in inflammatory status depending on the constitution of a certain hormonal status. Furthermore, menopause can lead to certain neuropsychiatric changes.

No information is provided regarding the presence or absence of topical and/or systemic treatment (corticosteroid therapy, biological therapy). The side effects of these treatments cannot be ignored.

  It is difficult to understand how the results of the questionnaires can be quantified correctly and objectively. Therefore, similar to other studies published in the literature, this research cannot pertinently indicate the causal relationship between atopic dermatitis and sexual dysfunction.

In the Results chapter, Fig. 1 shows the impact of atopic dermatitis on quality of life. Based on the general purpose of the study, the synergy between atopic dermatitis and depression on sexuality (female and male) should be highlighted.

Author Response

Reviewer:

Starting from the ultrastructural aspect of keratinocyte epithelial cells, intestinal epithelial cells, and the blood-brain barrier, the presence of the same tight apical junctional pattern, formed by membrane glycoproteins such as occludins and claudins, is highlighted. The expression of these molecules is controlled by a series of endogenous and exogenous factors, with repercussions on systemic homeostasis. Moreover, this morphological feature justifies and supports the involvement of the brain-skin-gut axis in the etiopathogenesis and clinical evolution of atopic dermatitis, one of the most common chronic immune-mediated inflammatory skin diseases.

To date, it is unclear whether anxiety and depression can be "unmasked" using skin keratinocytes or whether keratinocyte damage can attract and amplify depression/anxiety. Given the individual and social psycho-emotional impact of this pathology, the concern of the authors of this research is commendable. Therefore, the Introduction chapter could be supplemented with a table indicating the role of the main predisposing and/or contributing factors to sexual dysfunction in the context of atopic dermatitis.

Completing and using self-report questionnaires requires careful and thorough training by health professionals. Therefore, the Materials and Methods section must be completed with clear information on how to apply the various instruments used.

Reviewer: There are several degrees of severity of depression, which cannot be accurately assessed using a scale alone. Several scales are proposed in clinical practice—Hamilton, Beck, etc. On the other hand, the certification of data and the correct and adapted interpretation of results require the intervention of a specialist (psychologist and/or psychiatrist). It is not clear from the article whether this study benefited from the intervention of such a specialist.

Authors: We fully agree with the reviewer that depression cannot be diagnosed based only on a screening instrument such as HADS. However, HADS is a validated instrument to just assess depressive symptoms. We did not include a psychiatrist in our study, as the depression level was not the major aim of the study; we have concentrated on sexual dysfunction, and the level of depression was only an external measure to analyze the degree of sexual dysfunction and its influence on the patient's mood. 

Reviewer: Patients with gynecological/urological pathologies were excluded, but nothing is specified about the presence/absence of other chronic immune-mediated inflammatory pathologies such as autoimmune thyroiditis. It is known that females experience variations in inflammatory status depending on the constitution of a certain hormonal status. Furthermore, menopause can lead to certain neuropsychiatric changes.

Authors: Patients with known autoimmune diseases were excluded as well. However, we did not specifically test our patients against thyroid gland dysfunction, but the problem should be equally present in both the AD patients and the control group. 

Reviewer: No information is provided regarding the presence or absence of topical and/or systemic treatment (corticosteroid therapy, biological therapy). The side effects of these treatments cannot be ignored.

Authors: We thank the reviewer for this comment. We agree that some drugs may cause sexual disturbances. The patients had to be at least two and four weeks from topical and systemic treatment, respectively, before entering the study. However, we did not look for the treatment that was used by the patients in the past. 

Reviewer:  It is difficult to understand how the results of the questionnaires can be quantified correctly and objectively. Therefore, similar to other studies published in the literature, this research cannot pertinently indicate the causal relationship between atopic dermatitis and sexual dysfunction.

Authors: We agree with the reviewer on this statement. Thu, we have added this comment as the limitation of the study. 

Reviewer: In the Results chapter, Fig. 1 shows the impact of atopic dermatitis on quality of life. Based on the general purpose of the study, the synergy between atopic dermatitis and depression on sexuality (female and male) should be highlighted.

Authors: We agree with the reviewer that, considering sexuality, it is difficult to separate different aspects (such as AD and depression) from the sexual dysfunction. However, we cannot exclude the possibility that patients are more depressed because of sexual dysfunction. 

Round 2

Reviewer 1 Report

Comments and Suggestions for Authors

The authors have successfully addressed all my comments.

Author Response

The reviewer stated "The authors have successfully addressed all my comments." 

According to the above statement, we understand that no further changes are needed. However, if anything needs to be modified, we are happy to further revise our manuscript. 

Sincerely,

Adam Reich 

Reviewer 3 Report

Comments and Suggestions for Authors

The authors did not respond and did not complete the article in an appropriate scientific manner. The recommendations and information requested were not clarified. Therefore, I believe that the article can be resubmitted after careful and thorough revision.

Author Response

The Reviewer stated that "the authors did not respond and did not complete the article in an appropriate scientific manner. The recommendations and information requested were not clarified. Therefore, I believe that the article can be resubmitted after careful and thorough revision."

As no new criticism has been raised during the second round of the review process, we have again analyzed the comments provided during the first round of the review. Below we have included our answers:

Reviewer: 

Starting from the ultrastructural aspect of keratinocyte epithelial cells, intestinal epithelial cells, and the blood-brain barrier, the presence of the same tight apical junctional pattern, formed by membrane glycoproteins such as occludins and claudins, is highlighted. The expression of these molecules is controlled by a series of endogenous and exogenous factors, with repercussions on systemic homeostasis. Moreover, this morphological feature justifies and supports the involvement of the brain-skin-gut axis in the etiopathogenesis and clinical evolution of atopic dermatitis, one of the most common chronic immune-mediated inflammatory skin diseases.

Authors: We do believe that this comment does not consider our manuscript, as we did not analyze the role of the brain-skin-gut axis in the pathogenesis of atopic dermatitis. Otherwise, we would like to ask for a more precise comment.

Reviewer: 

To date, it is unclear whether anxiety and depression can be "unmasked" using skin keratinocytes or whether keratinocyte damage can attract and amplify depression/anxiety. Given the individual and social psycho-emotional impact of this pathology, the concern of the authors of this research is commendable. Therefore, the Introduction chapter could be supplemented with a table indicating the role of the main predisposing and/or contributing factors to sexual dysfunction in the context of atopic dermatitis.

Authors: Although we cannot exclude the possibility that keratinocyte damage/skin pathology may directly cause or influence depression and anxiety (e.g. by CRH production or some opioid receptor agonists), we do assume that this influence is more related to the feeling of stigma, subjective symptoms (itching, skin pain), low self-confidence because of visibility of skin lesions, etc. We think that providing a table that indicates the predisposing factors to sexual dysfunction exceeds the aim of the introduction, as it better fits the review paper (it may include a wide range of organic causes, psychogenic ones, but also some aspect of the pathogenesis of atopic dermatitis, which could be difficult to understand without a detailed explanation and an in-depth literature review. 

Reviewer: 

Completing and using self-report questionnaires requires careful and thorough training by health professionals. Therefore, the Materials and Methods section must be completed with clear information on how to apply the various instruments used.

Authors: We have provided the information that all participants were instructed before completing the questionnaires. We fully agree with the reviewer that the correct completion of the self-administered tools is critical for the validity of the study.  However, we would also like to underline that the senior author of the study (AR) is a long-standing member of the European Society of Dermatology and Psychiatry and has extensive experience in this type of research. 

Reviewer:

There are several degrees of severity of depression, which cannot be accurately assessed using a scale alone. Several scales are proposed in clinical practice—Hamilton, Beck, etc. On the other hand, the certification of data and the correct and adapted interpretation of results require the intervention of a specialist (psychologist and/or psychiatrist). It is not clear from the article whether this study benefited from the intervention of such a specialist.

Authors: We fully agree with the reviewer that depression cannot be diagnosed based only on a screening instrument such as HADS. However, HADS is a validated instrument to assess depressive symptoms. We did not include a psychiatrist in our study, as the depression level was not the major aim of the study; we have concentrated on sexual dysfunction, and the level of depression was only an external measure to analyze the degree of sexual dysfunction and its influence on the patient's mood. However, we have added the lack of psychiatrists as a limitation of the study. 

Reviewer:

Patients with gynecological/urological pathologies were excluded, but nothing is specified about the presence/absence of other chronic immune-mediated inflammatory pathologies such as autoimmune thyroiditis. It is known that females experience variations in inflammatory status depending on the constitution of a certain hormonal status. Furthermore, menopause can lead to certain neuropsychiatric changes.

Authors: Patients with known autoimmune diseases were excluded as well. However, we did not specifically test our patients against thyroid gland dysfunction, but the problem should be equally present in both the AD patients and the control group. Furthermore, the control group was not different in terms of patients' age and we assume that the problem of menopause was similar between the groups. 

Reviewer: 

No information is provided regarding the presence or absence of topical and/or systemic treatment (corticosteroid therapy, biological therapy). The side effects of these treatments cannot be ignored.

Authors: We thank the reviewer for this comment. We agree that some drugs may cause sexual disturbances. The patients had to be at least two and four weeks from topical and systemic treatment, respectively, before entering the study (this is now indicated in the text). However, we did not look for the treatment that was used by the patients in the past. 

Reviewer: 

It is difficult to understand how the results of the questionnaires can be quantified correctly and objectively. Therefore, similar to other studies published in the literature, this research cannot pertinently indicate the causal relationship between atopic dermatitis and sexual dysfunction.

Authors: We agree with the reviewer on this statement. Thus, we added this comment as a limitation of the study. 

Reviewer:

In the Results chapter, Fig. 1 shows the impact of atopic dermatitis on quality of life. Based on the general purpose of the study, the synergy between atopic dermatitis and depression on sexuality (female and male) should be highlighted.

Authors: We agree with the reviewer that, considering sexuality, it is difficult to separate different aspects (such as AD and depression) from the sexual dysfunction. However, we cannot exclude the possibility that patients are more depressed because of sexual dysfunction. A short commentary was added to the text. 

I do hope that the current version of our manuscript will be found suitable for publication in Medicina.